# Vaccines Against Urban Epidemic Arboviruses: The State of the Art

**DOI:** 10.3390/v17030382

**Published:** 2025-03-06

**Authors:** Cláudio Antônio de Moura Pereira, Renata Pessôa Germano Mendes, Poliana Gomes da Silva, Elton José Ferreira Chaves, Lindomar José Pena

**Affiliations:** Laboratory of Virology and Experimental Therapy (Lavite), Department of Virology, Aggeu Magalhães Institute (IAM), Oswaldo Cruz Foundation (Fiocruz), 50670-420 Recife, Brazil; claudio.prepara0@gmail.com (C.A.d.M.P.); renatapgm01@gmail.com (R.P.G.M.); poligs250@gmail.com (P.G.d.S.); chavesejf@cbiotec.ufpb.br (E.J.F.C.)

**Keywords:** emerging arboviruses, vaccine development, public health, mosquito-borne infections

## Abstract

Arboviruses represent a contemporary global challenge, prompting coordinated efforts from health organizations and governments worldwide. Dengue, chikungunya, and Zika viruses have become endemic in the tropics, resulting in the so-called “triple arbovirus epidemic”. These viruses are transmitted typically through the bites of infected mosquitoes, especially *A. aegypti* and *A. albopictus*. These mosquito species are distributed across all continents and exhibit a high adaptive capacity in diverse environments. When combined with unplanned urbanization, uncontrolled population growth, and international travel—the so-called “triad of the modern world”—the maintenance and spread of these pathogens to new areas are favored. This review provides updated information on vaccine candidates targeting dengue, chikungunya, and Zika viruses. Additionally, we discuss the challenges, perspectives, and issues associated with their successful production, testing, and deployment within the context of public health.

## 1. Introduction

Arboviruses (arthropod-borne viruses) represent a major challenge to global health worldwide. As their name suggest, the transmission of these pathogens to humans occurs through the bite of an infected arthropod, such as ticks and mosquitoes, with the latter being particularly noteworthy [1]. Among the arboviruses, dengue (DENV), chikungunya (CHIKV), and Zika viruses (ZIKV) are of particular concern because of their wide distribution, pathogenicity, and lack of approved treatment and vaccines.

Distributed across all continents and highly adapted to reproduce in human habitats (Figure 1) [2,3], mosquitoes of the *Aedes* genus, predominantly *A. aegypti* and *A. albopictus* [4,5], are vectors of these diseases and pose a complex challenge to disease control. The distribution of these vectors in tropical, subtropical, and even temperate zones is favored by their opportunistic feeding, generation of a large number of offspring, and versatility in the use of urban and peri-urban habitats [6,7,8]. Also, more than half of the global population is distributed in regions at risk of exposure to urban arboviruses [4]. The high dissemination of these mosquitoes results in overlapping zones of incidence, increasing the risk of infections and even simultaneous outbreaks [9], as seen in Colombia [10], Brazil [11,12], Mexico [13], and Guatemala [14] in the past few years.

Recent global data indicate a significant rise in dengue cases worldwide. There have been over 7.6 million reported infections, with 3.4 million confirmed cases, 16,000 severe cases, and approximately 3000 deaths. CHIKV and ZIKV also show considerable case numbers but to a lesser extent, with around 250,000 and 7000 cases, respectively [17]. In the Americas, the Pan American Health Organization/World Health Organization (PAHO/WHO) describes a total of 3,125,386 cases of arbovirus infections, with 90% attributed to dengue (2,811,452), 8.8% to chikungunya (273,685), and 1.3% to Zika (40,249) [18].

These viruses circulate in most of the tropical world and in some temperate countries, with disease distribution overlapping the spread of the vectors. In certain areas, dual and triple co-infections occur, leading to difficult disease diagnosis, as their clinical signs (headaches, myalgia, nausea, arthralgia, rash, etc.) cannot be distinguished. This scenario is favored by poor sanitation in the affected countries, low or absent herd immunity, the widespread distribution of its vectors, and high population mobility [19]. Co-infections in humans, despite a rare event, can be caused by a single, multiply infected mosquito or through sequential transmission events of different arboviruses. Although *Aedes aegypti* exhibits high permissiveness to co-infections with more than one arbovirus when feeding with mice [20], the frequency of this occurrence in natural conditions is still poorly understood [20]. There are reports of co-infection by DENV and CHIKV in countries such as Madagascar, India, Angola, Gabon, Singapore, Tanzania, Thailand, Malaysia, Myanmar, Nigeria, Saint Martin, Sri Lanka, and Yemen [21], which corresponds to approximately 13% of the transmission territory of these viruses. Additionally, co-infections between DENV and ZIKV [22,23] and between CHIKV and ZIKV [24] have also been reported. More rarely, infections with all three arboviruses in two Nicaraguans [25] and a pregnant Colombian woman [26] have also reported. Co-infections between these arboviruses are reviewed elsewhere [27]. Another factor that plays a crucial role in the epidemiology of these viruses is vertical transmission in the mosquito vector, which can occur when the virus infects the egg during oviposition or when the focus of viral infection is the germinal tissues of female mosquitos [28,29].

Unplanned population growth, coupled with unplanned urbanization and the international transport of goods, people, and animals, drive the magnitude and frequency of arbovirus infections [30] (Figure 2). These factors contribute to an increase in the circulation of these pathogens between large urban centers and accelerate their geographical expansion and emergence into new regions [31]. The scenario is exacerbated by the ecological consequences of climate change on the population of the insect vectors and viral evolutionary dynamics in the mosquitos [8,32]. The prevalence of these diseases is directly related to unplanned city growth and sanitation deficiency, which take place particularly in the outskirts of big cities [33].

There are no approved antivirals against these diseases, and the available treatment is based on supportive care. Thus, for decades, the control of these diseases has relied mainly on strategies to combat their vectors and breeding sites. More recently, vaccines have been developed and are considered a key component of disease prevention, even though their availability is still limited [41]. Given the complex scenario related to emerging arboviruses, priority has been given to action, the improvement of surveillance, and sanitation for future epidemics. In March 2022, the World Health Organization (WHO) launched the Global Arbovirus Initiative, aiming to concentrate resources on risk monitoring, pandemic prevention, preparedness, detection, and response against these diseases. Although there are other strategies to combat these pathogens, few large-scale combinations rival the positive impact on human well-being and health of vaccines and immunization programs [42]. In this review, we have compiled the state-of-the-art vaccine strategies to combat three emerging arboviral diseases, as well as the pre-existent technology, their development status, and short-, medium-, and long-term prospects. The vaccines discussed are summarized below in Table 1.

## 2. Dengue Virus (DENV I–IV)

Dengue virus is the most important arboviral disease given its prevalence, virulence, and widespread distribution. Annually, around 50 million people are infected, resulting in approximately 22,000 deaths [43,44,45]. In endemic countries, most DENV cases are reported in vulnerable populations, like infants and young children [46,47]. In recent decades, the number of cases has risen, and there are outbreak reports in Africa, Southeast Asia, the Americas, and even Europe [48]. To date, the virus is endemic in over 100 countries worldwide [43,44,45]. DENV incidence has increased almost eight times since 2000 [49]. Several factors are associated with this escalation in dengue cases, including the expansion of the mosquito vector, climate changes with global warming and high rainfall, the impact of the COVID-19 pandemic on the health system, which affected surveillance systems, the humanitarian crisis in several parts of the world, and increased traveling and tourism to dengue-endemic areas [50].

Until the 1970’s, dengue mosquito vectors had been identified only in a few countries. Currently, dengue is found in more than 130 countries [51]. The disease outbreaks tend to be cyclical, occurring every three to five years, following seasonal patterns with hot and rainy months when the mosquitoes usually breed [51,52]. In 2023, the Americas recorded an acute increase in dengue cases. According to the most recent historical series (2000–2023), Brazil alone is responsible for 80% of the cases [49]. In the same year, simultaneous circulation of all four serotypes has been detected in Brazil, Colombia, Costa Rica, Guatemala, Honduras, Mexico, and Venezuela, while in Argentina, Panama, Peru, and Puerto Rico, the DENV1, DENV2, and DENV3 serotypes circulate. In Nicaragua, the serotypes DENV1, DENV3, and DENV4 are prevalent [48]. Despite the circulation of *Aedes aegypti*, Chile is the only country in South America without autochthonous transmission of these arboviruses [53]. Countries with autochthonous profiles, such as Brazil, demonstrate a change in the virus epidemiological profile and an increase in the number of severe and fatal cases [49]. From a molecular standpoint, DENV is an enveloped virus, approximately 50 nm in size, with a positive-sense, single-stranded RNA genome, which means it can be directly translated into proteins [43]. The translation of the viral genome results in a single long polypeptide protein that is then cut into ten secondary proteins: three structural (capsid [C], pre-membrane [prM], and envelope [E]) and seven non-structural (NS1, NS2A, NS2B, NS3, NS4A, NS4B, and NS5) (Figure 3A) [43,44].

The virus belongs to the *Flaviviridae* family and genus *Orthoflavivirus*. Antigenically, it is characterized by four distinct serotypes (DENV-1, DENV-2, DENV-3, and DENV-4). Each serotype can be further separated into two to six independent genotypes [43,44,45,57].

The mutual presence of all four serotypes circulating in the same area becomes a challenge in the development of vaccines and treatments because of the need to overcome partial cross-protective immunity and the possibility of antibody-dependent enhancement of disease [58]. The pre-existing immunity to other serotypes constitutes a risk factor for the development of dengue with warning signs and severe dengue fever, which can result in death [59].

Dengue is manifested by a large range of clinical signs and symptoms. It can start as a mild and self-limiting illness, but it can suddenly become a more severe form or even evolve to death. Secondary heterotypic infection has been linked to an increased risk of severe forms of dengue, which is associated with systemic vascular leak (plasma leak) during the critical stage of defervescence [60]. The disease consequences are related to circulatory failure and hemorrhagic manifestations that are usually accompanied by distinctive symptoms, such as abdominal pain, persistent vomiting, bleeding gums or nose, blood in urine, and severe organ involvement, such as liver or kidney failure [60].

The search for a DENV vaccine began right after the disease spread globally, around the latter half of the 20th century [61]. Over the years, the challenges to be overcome delayed the development of a vaccine [62]. The need to simultaneously induce heterotypic (cross-reactive antibodies) and homotypic (type-specific antibodies) immunity to each new DENV serotype is a difficult one [63,64]. The vaccine candidate platform needs to induce protection against all four serotypes and, ideally, long-term immunity [63]. Neutralizing antibodies have been shown to correlate with protection, but the type of antibody is key to achieving protection.

It has been demonstrated that natural DENV infection produces protective neutralizing antibodies that recognize type-specific epitopes to each serotype. In contrast, an approved tetravalent vaccine (Dengvaxia) induced different neutralizing antibody subpopulations that recognized more conserved, cross-reactive epitopes between serotypes, suggesting that type-specific neutralizing antibodies are the best correlate of protection than total neutralizing antibody levels [65]. The ideal vaccine candidate should provide protective immunity against all four serotypes across all age groups, attend the delivery and cost implementation, and be protective in a single dose [66].

In the last few years, different types of strategies have been developed, from live attenuated vaccines to chimeric immunogens, purified–inactivated virus (PIV), recombinant subunit vaccines, DNA vaccines, vectored vaccines and virus-like particle (VLP)-based vaccines [67]. Currently, three dengue vaccine platforms have already been licensed or have reached Phase III in clinical trials.

### 2.1. Live Attenuated Vaccines

#### 2.1.1. CYD-TDV Dengvaxia^®^

A live attenuated vaccine named CYD-TDV (Dengvaxia^®^), developed by Sanofi Pasteur (Figure 4A), was first licensed in 2015 for use in Mexico, the Philippines, Brazil, and, later, in more than 20 countries. The vaccine is a live attenuated chimeric vaccine expressing DENV genes in the backbone of the yellow fever virus 17D vaccine strain. The used approach is a combination of the structural prM and E genes of all four DENV serotypes [68]. However, efficacy was limited by the DENV serotype, serostatus at vaccination, region, and age. Moreover, the vaccine is indicated only for use in DENV-seropositive individuals. CYD-TDV is not recommended for use in naïve populations due to an increased risk of severe DENV after infection because of antibody-dependent enhancement [69,70]. The vaccine induced more severe symptoms of DENV infection in seronegative subjects and children younger than 9 years upon exposure to wild-type DENV, which limited its use only to high seroprevalence rate areas [71,72]. Overall, the vaccine has an efficacy of about 80% against the outcomes of hospitalization, virologically confirmed symptomatic dengue, and severe dengue [72,73]. In a recent study in Brazil, the Dengvaxia^®^ vaccine had a vaccine effectiveness of 71% in individuals with a documented history of dengue. Nevertheless, vaccination was not associated with a significant reduction in the overall dengue case risk in individuals without a history of dengue infection [73]. Since its deployment requires a three-dose regimen and pre-vaccination screening for DENV antibodies, this vaccine is currently not being widely used.

#### 2.1.2. Takeda’s QDENGA^®^

Takeda’s Qdenga^®^ vaccine platform (Figure 4A) is a wholly DENV-based live attenuated vaccine. DENV-2 is used as a genetic backbone for the four serotypes, which provides a tetravalent feature to the vaccine candidate [74]. This vaccine strain was produced by 53 serial passages of the DENV-2 Thai strain (DENV-2 16681) in primary dog kidney cells, which resulted in attenuating mutations mapped to the 5’UTR, NS1, and NS3 genes [75]. The vaccines for DENV-1, 3, and 4 were made by replacing the structural prM and E genes of DENV-2 with their respective DENV serotype-specific genes to make the chimeras. The resulting viruses displayed temperature-sensitive and small plaque phenotypes [76]. The overall conclusion from pooled Phase 2 and 3 trials conducted in several countries in individuals aged 4 to 60 years old is that TAK-003 is well tolerated, irrespective of age, gender, or baseline dengue serological status [77]. A recent Phase 3 clinical trial was performed with children and adults living in dengue-endemic areas and concluded that TAK-003 (DENVax) has an efficacy of over 70% during the first year after immunization regardless of the DENV baseline serostatus [78,79].

The results highlight the vaccine safety and efficacy to dengue-naïve individuals and also previously dengue-exposed subjects [79]. A more recent clinical trial brought results from a three-year efficacy and safety Phase 3 clinical trial performed in eight countries across Asia and Latin America. Two doses of the vaccine had a cumulative efficacy after three years of 62.0% against virologically confirmed dengue and 83.6% against hospitalized dengue [80]. Despite the great advances achieved in this trial, a booster dose may improve the vaccine efficacy over time [80]. The vaccine efficacy (virologically confirmed dengue) was demonstrated against all four serotypes in dengue-exposed participants, with efficacy estimates ranging from 52.3% against DENV-3 to 80.4% against DENV-2. However, the efficacy in the seronegative population was adequate for DENV-1 and DENV-2, but insufficient against DENV-3. The low incidence of DENV-4 precluded an assessment of the vaccine efficacy against this serotype [81].

#### 2.1.3. LATV TV003 and TV005 Vaccines

A live attenuated tetravalent vaccine (LATV) is another more promising vaccine strategy developed by the National Institute of Allergy and Infectious Diseases (NIAID/NIH). LATV TV003 and TV005 vaccines are based on four recombinant live attenuated DENV components (rDEN1Δ30, rDEN2/4Δ30, rDEN3Δ30/3Δ31, and rDEN4Δ30) (Figure 4A). The vaccines are based on a thirty-nucleotide (Δ30) deletion in the 3′ UTR of each DENV serotype (DENV-1,3, and 4). In addition to Δ30, the rDEN3D30/31 component includes an additional 31-nucleotide deletion located 55 nucleotides upstream of the Δ30 mutation (rDEN3Δ30/3Δ31), to adequately attenuate the DENV-3 virus. The DEN2 vaccine was generated (rDEN2/4Δ30) using a chimerization strategy in which the prM and E genes of the DEN2 were cloned into the backbone of the rDEN4Δ30 vaccine candidate [82,83]. In order to induce homotypic antibodies to each of the four DENV serotypes, a balanced infectivity profile for all four components of the LATV is key. In this regard, the chimera rDEN2/4Δ30 was the least infectious when combined into tetravalent formulations (TV003 vaccine) at the dose of 10^3^ PFU for each virus [84,85]. The formulation presented a robust and balanced immune response following a single-dose administration, inducing trivalent or tetravalent neutralizing antibody responses in the vast majority of participants after a single primary dose [86,87].

However, the need for a boosting regimen is still under evaluation mainly for those subjects with pre-existing immunity [86,87,88]. The Brazilian Butantan Institute has partnered with the NIH to produce and test the vaccine (Butantan-DV) in an endemic population with significant pre-existing immunity against DENV infections [89]. The vaccine candidate is currently in Phase 3 clinical studies. In the two-year follow-up period, a single dose of Butantan-DV prevented symptomatic DENV-1 and DENV-2, regardless of dengue serostatus at baseline, with efficacies of 89.5% and 69.6%, respectively. Cases of DENV-3 or DENV-4 were not detected during the study period, and the efficacy against these serotypes could not be determined [90] The single-dose regimen reduces the vaccine cost per person and facilitates logistics and vaccination adherence, which are important factors for vaccine deployment in large populations. 

### 2.2. Subunit Vaccines

#### V180

V180 is a recombinant subunit vaccine expressing each of the four dengue serotypes in the Drosophila S2 expression system, being first tested in humans for its safety and immunogenic profiles. The developed platform is based on a truncated dengue E protein (DEN-80E) for all four serotypes and was tested in a Phase 1 randomized clinical trial in a flavivirus-naïve adult population in Australia. All nine different V180 formulations were produced with the ISCOMATRIX™ adjuvant, aluminum-hydroxide adjuvant, or without an adjuvant. The six V180 formulations with the ISCOMATRIX™ adjuvant were highly immunogenic after three vaccines doses at 1-month intervals, whereas formulations with the aluminum adjuvant (*n* = 1) and unadjuvanted formulations (*n* = 2) showed limited immunogenicity. The V180 formulations were generally well tolerated, despite a decrease in antibody levels six months after vaccination. All formulations adjuvanted with ISCOMATRIX™ (*n* = 6) demonstrated strong immunogenicity, whereas three formulations, one adjuvanted with aluminum and two unadjuvanted, were poorly immunogenic [91].

### 2.3. Inactivated Virus

#### Dengue Purified–Inactivated Vaccine (DPIV)

A Vero-cell-grown, formalin-inactivated tetravalent dengue vaccine has been developed and tested in clinical trials for its safety and immunogenicity. The dengue purified–inactivated vaccine (DPIV) formulations were adjuvanted either with aluminum hydroxide, AS01E, or AS03B at specific dosages. The formulations were tested in an adult population from a dengue-endemic area in Puerto Rico and followed-up for its safety and immunogenicity in a Phase 1, open-label clinical trial. The vaccine was tested through a two-dose regimen given intramuscularly on days 0 and 28 to flavivirus-naïve healthy adult volunteers. During the 3-year follow-up, all formulations were found to be safe and immunogenic. The levels of neutralizing antibodies against DENV 1–4 waned after 14 months but remained higher than pre-vaccination levels for all DENV serotypes, except for DENV-4. All formulations appeared to be safe and immunogenic during the 3-year follow-up [92].

### 2.4. DNA Vaccines

#### 2.4.1. D1ME100

A DNA vaccine was initially tested in mice with a vaccine formulation expressing either the truncated or the full-length DENV-1 E protein alone or in association with prM protein, as prM is necessary for the maintenance of the E protein conformation, an antigen whose neutralizing epitopes appear to be conformational. The vaccine expressing full-length E protein associated with prM was found to be more effective at inducing anti-dengue antibodies in a mouse model, and therefore, follow-up studies were conducted with this immunogen in monkeys. Two doses of the prototype DENV-1 vaccine provided 80–95% protection against a wild-type virus challenge in Rhesus macaques and Aotus monkeys. An open-label, Phase 1 clinical trial in humans given three intramuscular injections (0, 1, and 5 months) showed that this vaccine was safe but elicited neutralizing antibody responses in only 5 of 12 (41.6%) subjects in the high-dose (5.0 mg) group, showing that it is poorly immunogenic in humans [93].

#### 2.4.2. pNS1/E/D2

Recently, a DNA-based vaccine was made by expressing the DENV-2 E protein and NS1 in a bicistronic plasmid (pNS1/E/D2). Two doses of the vaccine elicited antibodies in mice and resulted in complete protection against a homologous wild-type DENV challenge [94].

#### 2.4.3. TVDV with Vaxfectin^®^

An open-label, Phase 1 clinical trial of a tetravalent dengue DNA vaccine (TVDV) formulated in Vaxfectin^®^, a cationic lipid-based adjuvant for DNA- and protein-based vaccines, was conducted in flavivirus-naïve volunteers to assess its safety and immunogenicity. The vaccine was safe and well tolerated but did not elicit a strong antibody response despite good induction of anti-dengue T-cell IFNγ responses in a dose-dependent manner [95]. Overall, DNA-based vaccines for dengue are well tolerated and safe, but they are hampered by their low immunogenicity.

### 2.5. RNA Vaccines

RNA vaccines have emerged as a safe and effective vaccine platform, especially after the COVID-19 pandemic when they were first approved for human use. Attempts to develop RNA-based DENV vaccines are underway. In one study, the authors evaluated an RNA vaccine expressing the most immunogenic epitopes of NS3, NS4B, and NS5 in mice. The vaccine induced a robust CD8^+^ T-cell immune response and protective immunity after being challenged with DENV1 [96]. Later, Wollner et al. used a nucleotide-modified mRNA vaccine encoding the DENV-1 prM and E and structural proteins from DENV serotype 1 encapsulated in lipid nanoparticles (prM/E mRNA-LNPs) and tested it in murine models. This vaccine was safe and elicited immune responses comparable to live DENV infection, resulting in high levels of neutralizing antibodies and DENV-specific CD4^+^ and CD8^+^ T cells, as well as full protection from the challenge [97].

## 3. Chikungunya Virus (CHIKV)

Chikungunya virus (CHIKV) is an arbovirus belonging to the *Alphavirus* genus and the *Togaviridae* family. CHIKV was first isolated in 1953 from the serum of a patient infected during an outbreak of a debilitating arthritic disease in the Tanganyika region, which is now part of Tanzania [98]. The term “chikungunya” is derived from the local language and means “those who bend up”, referring to the characteristic posture of individuals affected by the disease [99].

Between the 1960s and 1990s, the virus was detected as the causative agent of sporadic epidemics in various regions of Africa and South Asia, with epidemic intervals ranging from 7 to 20 years [100]. However, in 2004, there was a reappearance in Kenya, and its spread reached the islands of the Indian Ocean and India, extending to the southwestern Asian region. This outbreak affected millions of people, and since then, CHIKV has established itself as a global pathogen, spreading first to the South American continent and then reaching more than 50 countries in 2013, including southern Europe [101]. The disease has been reported in over 100 countries, causing seasonal or sporadic outbreaks. Alarming trends have emerged, particularly in the Americas. Between 1 January and 4 March 2023, a total of 113,447 cases of chikungunya were reported in the Americas, including 51 deaths, representing a four-fold increase in cases and deaths compared to the same period in 2022 (21,887 cases, including 8 deaths) [102]. In 2024 (as of 8 November), approximately 480,000 CHIKV cases and 190 deaths have been reported worldwide, with most cases being reported by Brazil, Paraguay, Argentina, and Bolivia [103].

Based on their respective geographical origins, the virus, which has only one serotype, is currently classified as West African (Senegal and Nigeria), East-Central South African (ECSA), Asian, and the Indian Ocean Lineage (IOL genotype). The ECSA lineage is further subdivided into two clades: ECSA1, consisting entirely of ancestral CHIKV sequences, and ECSA2, which contains sequences from the Central African Republic, Cameroon, Gabon, and the Republic of Congo [104].

The CHIKV genome is composed of positive-sense RNA of approximately 11.8 kb, with a capped 5′ end and a poly(A) tail at the 3′ end. The genome contains two open reading frames (ORFs) that encode the four non-structural proteins (nsP1, nsP2, nsP3, and nsP4), which are important in the formation of the replicase complex that catalyzes the production of new viral RNA, as well as five structural proteins (C-E3-E2-6K-E1) (Figure 3C) [105]. The E1 and E2 structural proteins carry the main viral epitopes and play a crucial role in virus attachment and entry into target cells, where E2 is responsible for receptor binding and E1 for membrane fusion, making them important targets for therapy [106].

The virus circulates in two distinct transmission scenarios, with enzootic transmission involving a sylvatic cycle among non-human primates through arboreal *Aedes* spp. mosquitoes in sub-Saharan Africa and urban transmission occurring between humans, mainly through the bites of female mosquitoes of the *Aedes aegypti* and *Aedes albopictus* species [107]. The incubation period is relatively short and can range from 2 to 7 days. After this period, during the acute phase of the disease, the intensity of symptoms correlates with viremia, resulting in a characteristic clinical triad of CHIKV infection, including a rash (in 50% of cases), high fever, and intense myalgia and arthralgia. Arthralgia is usually symmetrical and tends to affect distal joints more than proximal ones [108]. Additional symptoms include nausea, fatigue, headache, and back pain. Although rare, neurological complications of the disease are the most concerning, given their higher association with intensive care unit (ICU) admissions and mortality. CHIKV is known to affect the central nervous system, causing a wide range of symptoms, including encephalitis and Guillain–Barré syndrome [109].

While most infections are asymptomatic or self-limiting, with most symptoms disappearing within 7 to 10 days, many patients report prolonged arthralgia that can last for weeks, months, or even years, characterizing the chronic phase of the disease. As it progresses to the chronic form, the patient develops persistent chronic polyarthralgia, a debilitating condition that affects not only the patient’s mobility but also their well-being and quality of life, resulting in an inability to perform daily activities [110]. It is estimated that 40–80% of patients progress to the chronic form when immune responses are implicated, leading to an increase in inflammatory cytokine IL-6 and IL-17 levels more frequently than in patients with an ongoing direct infection [111]. The fatality rate is estimated at 0.3–1 per 1000, with most deaths reported in newborns, adults with underlying conditions, and the elderly [112]. The severity of CHIKV infection and long-term health effects combined with its wide distribution underlines the pressing need for specific treatments. Presently, despite its endemic presence in many regions, there exists no licensed targeted therapy for acute CHIKV infection. Treatment primarily revolves around supportive measures, such as pain relievers, anti-inflammatories, rehydration, and rest [113].

The early 2000s witnessed a surge in the interest to develop a vaccine against CHIKV due to its rapid spread across the Indian Ocean and Southwest Asia. Multiple technological platforms were explored, leading to the development of several experimental vaccine candidates that are currently in various stages of development. Notable CHIKV vaccine candidates undergoing clinical development encompass inactivated virus vaccines, virus-like particle vaccines, as well as DNA and RNA vaccines. These promising advancements in vaccine research hold the potential to mitigate the impact of CHIKV and reduce its burden on affected populations [114].

### 3.1. Inactivated Virus Vaccinesplu

Throughout history, several attempts have been made to develop an effective CHIKV vaccine. One of the earliest recorded efforts in the 1960s involved experimental vaccines with African CHIKV strain 168, which were formalin inactivated. These vaccines were developed at the U.S. Army Medical Research Institute of Infectious Diseases (USAMRIID) and produced from African green monkey kidney tissue and chicken embryo cells. The former induced good immune responses, but the latter was poorly immunogenic [115]. The formalin-inactivated CHIKV vaccine grown in monkey kidney cells was safe and efficacious in pre-clinical tests using mice and Rhesus macaques [116]. A few years later, in 1971, this vaccine was developed using an Asian strain and tested in human volunteers. The complete lack of adverse reactions or side effects, along with the strong immune response observed in volunteers, confirmed the safety and immunogenicity of this vaccine [117]. More recently, the BBV87 candidate, consisting of a whole virus inactivated by beta-propiolactone, has entered Phase 2 and 3 clinical trials in humans. The vaccine is formulated with 0.25 mg of aluminum hydroxide and administered intramuscularly in 0.5 mL doses. The trials involved single and escalating doses (10–40 µg), demonstrating immunogenicity, safety, and good tolerability. More details on the candidate can be found at NCT04566484 (also available in Table 1).

### 3.2. Live Attenuated Virus Vaccines

#### 3.2.1. 181/Clone 25 (181/25 or TSI-GSD-218)

Live attenuated virus vaccines offer effective and long-lasting immunity, require less frequent administration, and have lower production costs compared to inactivated vaccines. The first live attenuated CHIKV vaccine to advance into clinical trials was TSI-GSD-218, also known as strain 181/clone 25 (Figure 4C). This candidate live attenuated vaccine was also developed at the USAMRIID and manufactured at The Salk Institute-Government Services Division (TSI-GSD) for clinical trials. The Thai strain isolate AF5561 was continuously passed in renal cells and human embryonic cells (MRC-5) until it achieved significant attenuation. The vaccine showed strong immunogenicity in animal models; however, despite its promising prospects, during Phase 2 trials, the vaccine caused mild and transient arthralgia in 8% of the vaccinated individuals, and the virus isolated from their blood showed a reversion of its attenuating mutations [118].

#### 3.2.2. CHIKV/IRES Vaccine

The CHIKV/IRES vaccine (Figure 4C) represents a significant breakthrough in the quest for an effective vaccine against CHIKV. This vaccine was developed using an attenuation strategy based on the Internal Ribosome Entry Site (IRES) of the encephalomyocarditis virus (EMCV), which had previously been tested in the TC-83 vaccine. The IRES element was used to replace the sub-genomic promoter of CHIKV, thereby controlling the expression of the virus’s structural proteins and reducing its replication. The results demonstrated that CHIKV/IRES replicated more slowly compared to wild-type CHIKV and the 181/25 vaccine strain. Furthermore, CHIK/IRES was found to be incapable of infecting vector mosquitoes, effectively preventing transmission back to the mosquito population.

The safety and efficacy of the CHIK/IRES vaccine were evaluated in mouse and non-human primate (NHP) models. These studies showed that the vaccine induced a robust immune response with the production of neutralizing antibodies, even after a single dose, providing protection against a lethal virus challenge. Currently, CHIKV/IRES vaccine candidate is progressing to clinical trials and is in Phase 1 testing by the manufacturer, Takeda [119].

#### 3.2.3. VLA1553

The VLA1553 vaccine, developed by Valneva (Figure 4C), aimed at providing comprehensive protection against all circulating CHIKV strains with a single dose. To achieve this goal, the vaccine was derived from the LR2006 OPY1 strain, which belongs to the ECSA genotype. The virus used in the vaccine has a deletion of 61 amino acids in the nsP3, an essential gene for CHIKV replication. This genetic modification is responsible for attenuating the virus in vivo, making it safe for human use.

The candidate has been tested in Phase 3 of a clinical trial, a critical stage that evaluates its effectiveness and safety in a large population. The vaccine demonstrated an excellent immunogenicity profile, with 98.9% of participants developing a positive serological response after receiving a single dose of the vaccine. Furthermore, this immune response remained robust over time, with 96.3% of vaccinated individuals maintaining protective levels of neutralizing antibodies against CHIKV for up to 180 days after receiving the vaccine [120]. The vaccine is generally well tolerated, although some side effects, such as headache, fever, arthralgia, and myalgia, have been noted in some patients [121]. Following a single VLA1553 vaccination, chikungunya virus-neutralizing antibodies remained above the protective threshold for up to 2 years, with no long-term serious adverse events associated with the vaccination [122].

In November 2023, the VLA1553 vaccine, whose trade name is Ixchiq^®^, was approved by the FDA in the USA, as well as in Canada and Europe in June 2024. Regulatory evaluations for IXCHIQ^®^ are currently in progress in the United Kingdom and Brazil. Additionally, a clinical trial in adolescents, VLA1553-321, is underway in Brazil. 

Despite these promising results, it is important to highlight that the protective efficacy of VLA1553 in endemic areas is still being rigorously investigated. Ongoing research aims to provide a more in-depth understanding of the vaccine’s performance in real-world scenarios, where exposure to the virus is more common.

### 3.3. Virus-like Particles (VLPs)

#### VRC-CHKVLP059-00-VP (PXVX0317)

Virus-like particles (VLPs) have emerged as a significant advancement in the development of vaccines. One of the early vaccine candidates that made notable progress is based on virus-like particles (VLPs). These VLPs, termed VRC-CHKVLP059-00-VP, were generated from the structural polyprotein genes (capsid, E3, E2, 6K, and E1) derived from the Senegal 37997 CHIKV strain. The CHIKV VLP structure closely resembles that of wild-type virions, containing all structural proteins, but crucially, it does not pose any risk of viral replication, as no viral genetic material is incorporated. The VLP-based CHIKV vaccine, without the use of adjuvants, underwent initial evaluation in a non-human primate model, resulting in the production of high-titer neutralizing antibodies that neutralized heterologous strains. Importantly, the vaccine effectively controlled viral replication and dissemination after a CHIKV challenge, and there were no reports of severe adverse events related to the vaccine [123]. The vaccine was then GMP manufactured via the transfection of VRC293 cells (suspension-adapted, serum-free HEK293 cells) with plasmid DNA expressing the CHIKV structural genes and tested in humans. A Phase 1 clinical trial in humans showed that VRC-CHKVLP059-00-VP was highly immunogenic after a single dose while also being safe and well tolerated [124].

Also known as PXVX0317 when adjuvanted with aluminum hydroxide, this VLP vaccine was tested in a Phase 2 clinical trial to assess its safety and immunogenicity across various doses, vaccination schedules, and formulations, including its co-administration with aluminum hydroxide as an adjuvant. PXVX0317 was well tolerated and triggered a strong, long-lasting, serum-neutralizing antibody response against CHIKV for up to 2 years. The presence of the adjuvant in the vaccine preparation enhanced the antibody response after a single dose [125]. A single 40 μg dose of adjuvanted PXVX0317 is currently being tested in Phase 3 clinical trials (NCT05072080 and NCT05349617).

### 3.4. Viral Vector Vaccines

#### 3.4.1. MV-CHIK

MV-CHIKV is a recombinant-based live attenuated vaccine, using the measles vector derived from the Schwarz vaccine strain. This measles vector has been modified to become a highly versatile recombinant vector that expresses CHIKV structural genes (structural genes C, E3, E2, 6K, and E1). The choice of the Schwarz strain was based on its commercial availability and approval by the World Health Organization (WHO), as well as its recognition as one of the safest, most effective, and widely used measles vaccines in the world.

In 2015, the first Phase 1 study in humans was conducted, revealing promising results in terms of immunogenicity after one or two doses of the vaccine, as well as a safety and tolerability profile that proved to be acceptable. In the Phase 2 study, a single dose of the vaccine was able to stimulate the production of neutralizing antibodies in a range that varied from 50% to 93% of participants, and after a second dose, these levels increased significantly, reaching between 86% and 100% of immune response in all treatment groups within a maximum of 6 months after the booster vaccination [126].

#### 3.4.2. ChAdOxI-Chik/CHIK001

The candidate vaccine ChAdOx1-Chik is a chimpanzee adenoviral vector with replication deficiency, known as ChAdOx1, initially developed by the University of Oxford (UK). The same vaccine platform has been approved and marketed for human use against COVID-19 and became known as the Oxford–AstraZeneca COVID-19 vaccine. Despite its efficacy, the vaccine was associated with an elevated risk of rare to but potentially fatal thrombosis with thrombocytopenia syndrome (TTS), which has been linked primarily to younger female recipients of the vaccine [127]. The vaccine has been withdrawn from the global market in 2024 because of low demand, according to the manufacturer company [128].

The ChAdOx1-Chik vaccine contains the genes of the CHIKV that encode the structural proteins of the virus, resulting in the production of virus-like particles (VLPs) from various CHIKV lineages, including the Asian, ECSA, and West African lineages. The safety and immunogenicity of the vaccine were evaluated in a Phase 1 clinical trial that included healthy adults. When administered as a single dose, the ChAdOx1-Chik vaccine proved to be safe and well tolerated. It also induced the production of neutralizing antibodies against all four virus lineages in all participants, two weeks after vaccination, at all tested doses. No serious adverse reactions were observed [129]. Currently, this platform is undergoing Phase 1b testing to assess safety and immunogenicity when administered alone or in co-administration with the ChAdOx1-ZIKV vaccine in healthy adults aged 18 to 50, residing in the Monterrey metropolitan area in Mexico. The study was concluded in March 2022, but as of now, the results have not been published.

#### 3.4.3. VSVΔG-CHIKV

A vesicular stomatitis virus (VSV)-based vaccine vector expressing the complete CHIKV precursor polyprotein E3-E2-6K-E1 has been evaluated in pre-clinical studies. To achieve this, two plasmid constructs were created: one with the complete VSV inserted with the CHIKV polyprotein and another similar one but with the gene for the VSV natural glycoprotein G, which is responsible for the infectivity of the virus, completely removed. This latter construct was named VSVΔG. Surprisingly, the authors observed that the chimeric vector VSVΔG expressed and efficiently incorporated the CHIKV glycoproteins into the viral envelope, replacing the position where the original glycoprotein G would have been, and was able to propagate without the need for it.

To assess the ability of the recombinant VSV vectors to induce immune responses, C57BL/6 mice were immunized with 10^6^ plaque-forming units (PFU) of the VSV vectors via intramuscular injection. The mice were then challenged with the wild-type CHIKV-LR (La Reunion) strain, receiving 10^4^ PFU per mouse via subcutaneous injection, and monitored for 10 days for signs of infection. Both chimeric vectors, with or without the VSV glycoprotein G, induced neutralizing antibody (nAb) responses to CHIKV after a single dose, with the VSVΔG generating a stronger nAb response compared to the full chimeric vector [130].

#### 3.4.4. MVA-CHIK

The MVA-CHIK candidate is based on a highly attenuated vaccine strain of the modified Ankara virus (MVA), an attenuated strain of the vaccinia virus, which is safe in humans and induces a strong immune response [131]. The vaccine, which expresses the CHIKV E2 and E3 proteins, has been tested in murine models using BALB/c mice (immunocompetent) and A129 mice (deficient in IFNα/β). A consistent protection against viremia and joint swelling, critical markers of CHIKV infection, was observed. Furthermore, the candidate provided 80% protection against mortality in interferon-knockout mice just 11 days post-immunization. Despite eliciting immune responses, the detection of neutralizing antibodies was low or absent. However, transferring immune serum from vaccinated animals to naïve mice did not prevent death, suggesting that antibodies might not be the primary mediators of protection provided by the vaccine. Additionally, removing CD4^+^ T cells (but not CD8^+^ T cells) from vaccinated mice resulted in 100% mortality, indicating that CD4^+^ T cells play a crucial role in the protection offered by MVA-CHIK [132].

### 3.5. mRNA Vaccines 

#### mRNA-1388 (VAL-181388) Vaccine

The mRNA-1388 (VAL-181388) vaccine, developed by Moderna, encodes the entire CHIKV structural polyprotein, which includes the capsid and envelope proteins E3, E2, 6k/TF, and E1, derived from the West African genotype. The mRNA is delivered via a lipid nanoparticle encapsulation system. The mRNA-based vaccine candidate was tested for immunogenicity and safety in a Phase 1 trial involving healthy adults aged 18 to 49 residing in a non-endemic region for CHIKV. Conducted in the United States from July 2017 to March 2019, the study was placebo controlled, randomized, and included dose variation, with 90% of participants completing the trial. All administered dose variations induced favorable safety profiles and robust humoral responses, with an increase in dose-dependent neutralizing antibodies. The geometric mean titers (GMTs) 28 days post-immunization were as follows: 6.2 for 25 μg; 53.8 for 50 μg; 92.8 for 100 μg; and 5.0 for the placebo group. Persistent humoral responses were observed up to one year after the trial in the two higher-dose groups, with a similar trend in neutralizing antibodies and CHIKV-binding antibodies. In addition to being well tolerated, the mRNA-1388 candidate elicited substantial and lasting (up to 1 year) neutralizing antibody responses in participants [133].

## 4. Zika Virus (ZIKV)

Zika virus (ZIKV) was first isolated in 1947 in the Zika Forest, Uganda [134,135]. For many years, the virus has circulated only in the African continent. Then, the virus moved from Africa and Asia to the Pacific Island of Yap in Micronesia in 2007 and to additional Pacific islands, including French Polynesia in 2013. Between March 2015 and the end of January 2016, over 20 countries reported outbreaks of this viral type [136]. In November 2015, a surge in microcephaly cases in Brazil was associated with the ZIKV epidemic, bringing global public health attention to it [111]. In February 2016, the WHO classified the disease as emerging and of great concern to global public health due to its rapid and uncontrolled spread. Currently, the virus is considered endemic in 79 territories and continues to circulate in the tropical and subtropical regions of over 87 countries [137]. According to data from the epidemiological bulletin on the disease in Brazil (2015–2023), 22,251 suspected cases were reported, of which 3742 (16.8%) tested positive for some congenital infection. Among these, 1828 (48.9%) were classified as Congenital Zika Syndrome (CZS), with 1380 (75.5%) recorded in the Northeast region of the country. Of the total confirmed CZS cases, 261 (14.3%) resulted in death. Additionally, 2877 (12.9%) cases remain under investigation. The Brazilian states with the highest number of cases were Tocantins (530), São Paulo (344), Espírito Santo (237), Rio Grande do Norte (229), and Rio de Janeiro (220) [138].

In adults, ZIKV infection typically presents mild and non-lethal symptoms in 20–25% of cases, with most infections being clinically asymptomatic in the initial phase [139]. When present, symptoms resemble those of the common flu, accompanied by other signs such as low and variable fever, rash conjunctivitis, arthritis, or arthralgia [140]. Evidence indicates ZIKV as the primary etiological agent involved in the genesis of microcephaly andother congenital malformations such as hearing, visual, and neuropsychomotor alterations in newborns during the 2015–2016 epidemics in the Americas [141,142].

The virus replication in embryonic cells is quite peculiar, showing a clear ability to cause damage and a preference for infecting developing neural cells. One potential cause of this preference may be the tissue-specific expression of specific cellular receptors, a line of thought that still lacks approaches for better understanding [139,143]. While considered a transient infection in adult individuals, evidence suggests that the consequences of ZIKV infection may also impact neural stem cells in adult individuals [144].

Similar to other flaviviruses, the genome of the ZIKV consists of a single-stranded, positive-sense RNA molecule of 10.8 kb, encoding a single polyprotein processed into three structural proteins—capsid (C), pre-membrane (prM), and viral envelope protein (E)—along with seven other non-structural proteins (NS1, NS2A, NS2B, NS3, NS4A, and NS4B) (Figure 3B) [145].

The virus has become endemic, and it is considered a global health and socioeconomic threat. Despite this, there are no prophylactic treatments or vaccines to combat the infection. Various technologies have been employed as a foundation for developing vaccines against ZIKV, such as attenuated viruses, vectored vaccines, viral vector vaccines, and modified RNA or DNA. Regardless of the available technology, the landscape of combating ZIKV is as challenging as that observed in other emerging arboviruses, with a notable concern about its potential for enhancing DENV infection and ability to cause Guillain–Barré syndrome [146].

### 4.1. Live Attenuated Vaccines

#### 4.1.1. ZIKV-10-del

ZIKV-10-del is a vaccine candidate that carries a deletion of ten nucleotides in the 3′ untranslated region of the original ZIKV genome (Figure 4B). A single dose of the candidate induced sterilizing immunity and completely protected A129 mice challenged with live ZIKV. Additionally, a robust T-cell response was also observed. In an intracranial inoculation assay in CD-1 mice with 1 × 10^4^ PFU of ZIKV-10-del, no deaths occurred, in contrast to the lethality observed in mice inoculated with 10 PFU of wild-type ZIKV. Furthermore, the approach demonstrated the inability of ZIKV-10-del to infect mosquitoes, enhancing the safety profile of the candidate. The live attenuated vaccine in question showed high immunogenicity and elicited sterilizing immunity against ZIKV infection [147].

#### 4.1.2. ChinZIKV

This vaccine candidate was developed based on the genetic foundation of an already licensed attenuated vaccine for Japanese encephalitis (JEV), the SA14-14-2. ChinZIKV is a chimera formed by replacing the prM-E gene of the JEV SA14-14-2 strain with a corresponding region from the Asian strain of ZIKV FSS13025 (Figure 4B). ChinZIKV was successfully rescued in BHK-21 cells and correctly expressed the structural proteins of the ZIKV and the non-structural proteins of the JEV, as confirmed via immunostaining and Western blotting. The viral chimera reached a maximum titer of 10^6^ PFU ml-1 in Vero cells. The vaccine study was conducted in BALB/c mice, A129 mice, and Rhesus macaques in a single dose, generating robust immune responses and providing complete protection against a ZIKV challenge. The candidate also conferred protection against placental damage resulting from the virus infection in females challenged during pregnancy [148].

#### 4.1.3. YF-ZIKprM/E

The chimeric vaccine candidate YF-ZIKprM/E was modified by replacing the regions encoding the surface glycoproteins prM-E of one of the safest yellow fever vaccines (YFV), the 17D [149], with the corresponding sequences from the ZIKV (Figure 4B). The vaccine study was conducted in AG129 mice, which are knocked out for interferon α/β and -γ, wild-type C57BL/6 mice, and mice knocked out for interferon α/β. The animals were intraperitoneally vaccinated with 1 × 10^4^ PFU of YF-ZIKprM/E, while the control group received a MEM solution supplemented with 2% fetal bovine serum. Following immunization, the animals were challenged with lethal doses of both ZIKV and YFV. The candidate provided complete protection to the animals against both viruses, demonstrating the vaccine’s potential as a dual candidate, protecting vaccinated animals against both ZIKV and yellow fever virus, regardless of whether the mice had knockout genes or not [150].

### 4.2. Whole Inactivated Virus

#### 4.2.1. ZPIV

The Zika Purified–Inactivated Virus vaccine (ZPIV) was produced based on a circulating strain in Puerto Rico (PRVABC59). The stock of this strain was generated in low-passage Vero cells infected with a multiplicity of infection (MOI) of 0.01 PFU per cell. The vaccine was a Vero-cell grown ZIKV inactivated by formalin. The immunogenicity of the candidate was tested in macaques. All vaccinated animals developed specific neutralizing antibodies against ZIKV and specific antibodies in the second week post-immunization, with median titers of 1.87 and 2.27, respectively. After the booster dose was administered in the fourth week, these titers increased to 3.54 and 3.66. No antibody response was detected in animals receiving a placebo. Subsequently, all animals were challenged with 10^3^ PFU of ZIKV-PR (Puerto Rican strain) or ZIKV-BR (Brazilian strain), with viral infectivity confirmed by growth in Vero cells and viral loads quantified via qRT-PCR. Monkeys in the control group exhibited detectable viremia from days 6 to 7, with a maximum viral load of 5.82 log copies/mL on days 3 and 5 post-challenge, contrasting with the observations in vaccinated individuals, who showed an absence of detectable virus (<100 copies/mL) in blood samples, colorectal secretions, urine, and cervicovaginal secretions. Plasma viral loads were indistinguishable for both ZIKV strains [151].

A recent study has shown that ZPIV was able to prevent vertical transmission of the virus in hSTAT2KI mice when administered prior to pregnancy, providing protection to the offspring for up to 28 days after birth. Additionally, both maternal and fetal infections were prevented through the use of human hyperimmune sera via the passive transfer of IgG antibodies [152]. The ZPIV was also effective in preventing the vertical transmission of ZIKV in marmosets when administered before pregnancy. During pregnancy, the vaccine induced neutralizing antibody responses similar to those observed in the pre-pregnancy model without causing adverse effects [153]. The candidate is being tested in human trials (NCT02952833, NCT02937233, NCT03008122, and NCT02963909), with promising results for subsequent evaluation phases.

#### 4.2.2. Zika Virus Vaccine MR766

Virus inoculation using this vaccine approach was performed in Vero cells at an MOI of 0.01 PFU/cell and collected 5–6 days post-inoculation. The viral material was clarified, filtered, and subsequently inactivated with formalin solution. The purified inactivated Zika virus antigen was produced with 0.25 mg of aluminum/dose. Two groups of female AG129 mice deficient in interferons I and II (4 to 6 weeks old, *n* = 8/group) were vaccinated with doses of 5 or 10 μg on days 0 and 21 intramuscularly. Subsequently, they were challenged subcutaneously with 10^4^ PFU of ZIKV FSS 13025 (Cambodian strain) or MR 766 (African strain) on day 28. All vaccinated mice were protected against both ZIKV strains. The placebo-treated animals showed gradual morbidity progressing to death, with a mean time to death (MTD) of 8 days for the MR 766 strain and 12 days for FS 13025. The mean antibody titers based on ZIKV as the coating antigen were 3.35 and 4.26 (*p* < 0.0001) in the groups receiving 5 μg of the vaccine and 3.50 and 4.41 (*p* < 0.0001) in those receiving 10 μg. The vaccine candidate also demonstrated efficacy in passive immunization of BALB/c mice via its antiserum, with no infectious viral particles after passive immunization, while animals receiving pre-immune serum showed a peak viremia at 72–96 h [154].

### 4.3. DNA Vaccine Candidates

#### 4.3.1. GLS-5700

The GLS-5700 candidate is a synthetic DNA vaccine designed for the expression of a consensus of pre-membrane and envelope antigens of the ZIKV. The safety and immunogenicity of the GLS-5700 candidate were evaluated in a Phase 1 clinical trial with two groups of 20 participants each, averaging 38 years of age, with 60% being women. The participants received 1 or 2 mg of the candidate intradermally followed by electroporation at the beginning, fourth, and twelfth weeks of the study. Local reactions were detected in approximately 50% of the participants, such as local pain, swelling, redness, and itching, without severe effects. The presence of binding antibodies was detected in all participants, with geometric mean titers (GMTs) of 1642 for 1 mg doses and 2871 for 2 mg doses. The candidate induced binding antibodies in 95% of participants after two doses and in 100% after three doses were administered. Furthermore, the immune sera retrieved from the study participants after vaccination also prevented the death of mice in an in vivo assay [155].

#### 4.3.2. VRC5283 and VRC5288

The vectors VRC5283 and VRC5288 were derived from a ZIKV strain isolated from French Polynesia (H/PF/2013) and were based on the expression of the structural proteins prM and E for the production and release of virus-like subviral particles (SVPs) from transfected cells. To enhance their expression, the VRC5283 vector had its original prM signal sequence replaced by the analogous region from the Japanese encephalitis virus (JEV). The prM region of JEV can positively influence the processes of virus particle assembly and maturation through the E protein [156]. This vector was further modified to create a subsequent construct, VRC5288, by exchanging the last 98 amino acids of the E gene, with the goal of improving SVP secretion. Both vectors were effectively expressed in mammalian cells, with electron microscopy analysis detecting rounded particles consistent with the appearance of other flavivirus-derived SVPs. The immunogenicity of each candidate was assessed in BALB/c and C57BL/6 mice via intramuscular (quadriceps) electroporation. Vaccination with both candidates induced ZIKV-neutralizing antibodies (NAbs) that remained similar even with varying administered doses.

The immunogenicity of the vectors was subsequently tested in a study with Rhesus macaques, administered intramuscularly using a needleless injection device (Pharmajet), with six animals per group receiving two doses of 1 mg of the VRC5283 vector or 4 mg (VRC5283 and VRC5288) at weeks 0 and 4, with one of the groups receiving a single dose of 1 mg of VRC5288. All evaluated individuals showed statistically higher immune responses of NAbs than macaques that received the control vector VRC8400 (*p* = 0.022), and it was possible to detect these neutralizing and binding antibodies reaching their peak in the third week. Individuals who received a single dose had lower NAb titers than those who received two doses of either vector [157].

### 4.4. mRNA Vaccines

#### 4.4.1. mRNA-1325

A modified mRNA of ZIKV was designed to encode the structural genes prM and E of ZIKV for the production of virus-like particles (VLPs) and encapsulated in lipid nanoparticles (LNPs) for intramuscular delivery. The efficacy of the candidate was tested in mice challenged with the 96–740 strain of ZIKV. Two doses of the encapsulated mRNA encoding the prM-E genes of ZIKV resulted in high titers of neutralizing antibodies (~1/100,000), providing not only protection, but also sterilizing immunity to the challenged animals [158].

#### 4.4.2. mRNA–LNP

The vaccine candidate is an mRNA encapsulated in lipid nanoparticles (LNPs) encoding the prM and E genes of ZIKV H/PF/2013, incorporating the modified nucleoside 1-methylpseudouridine (m1Ψ) in place of a uridine nucleotide. The immune response of the candidate was tested in mice and Rhesus macaques. In mice, the candidate induced multifunctional CD4^+^ T-cell responses specific to the E protein in the second week post-vaccination. At weeks 8–12, a rapid development of Zika virus-specific immunoglobulin G (IgG) was detected in the sera of vaccinated animals, with a final titer of 180,000 (90 μg mL^−1^). Mice challenged in the short and long term were also protected against detectable viremia, regardless of the challenge timing. The study conducted in macaques with different doses of the candidate indicated that the lowest administered dose (50 μg) was more than sufficient to generate immunogenicity against ZIKV infection [159].

### 4.5. Protein Vaccine

#### 4.5.1. Plant-Produced ZIKV E (PzE)

The development of this candidate was based on the ZIKV E protein. The immunogenicity was tested in mice. Biochemical approaches indicated that the plant-expressed ZIKV E protein (PzE) exhibited specific binding to monoclonal antibodies capable of recognizing PzE epitopes. The transient expression of the protein in question led to its accumulation in the leaves of wild tobacco plants (*Nicotiana benthamiana*). Mouse studies demonstrated that the candidate was highly immunogenic, inducing robust responses with two administered doses. Antibody titers indicated that the candidate-driven response exceeded the immunity threshold against different ZIKV strains. PzE emerged as a potential option for the production of vaccines that are not only safe but also affordable against ZIKV, as plant expression systems are a promising approach to reduce the cost of biologicals [160].

#### 4.5.2. E Protein from s2 Insect Cell

This vaccine candidate developed by the University of Hawaii is based on the expression of ZIKV E protein in Drosophila melanogaster fruit fly S2 cells. The protein was collected from the s2 cell supernatant and doubly purified, yielding a soluble protein of approximately 45 kDa with a purity above 90%. The candidate immunogenicity was tested in three mouse strains (Swiss Webster, BALB/C, and C57BL/6). Results from the vaccine trial suggested a significant production of ZIKV-specific antibodies titers when administered with clinically relevant adjuvants. Furthermore, full protection against ZIKV infection was achieved with two vaccine doses and was conferred by the production of neutralizing antibodies [161].

### 4.6. Viral Vectored Vaccines

#### 4.6.1. MV–Zika

A panel of live vectors based on the measles vaccine (MV) was engineered to express ZIKV E and NS1 proteins. The prM and E genes were inserted into the transcription cassette at two distinct positions: upstream of the N gene, forming the candidate VM-E0, and between the N and P genes, generating the candidate VM-E2. Vaccine challenges in mice indicated that both candidates provided protection against infections with the non-lethal Asian strain of ZIKV (PRVBC59) and the lethal African strain (MR766), resulting in 100% survival of the animals. However, mice immunized with the MV-E2 candidate still showed viral presence in the brain and reproductive organs when lethally challenged.

Two additional candidates capable of expressing the ZIKV E2 and NS1 proteins were also tested, either separately or in combination with the antigens. The vaccine containing the dual antigen (E2 + NS1) protected mice from infection by the African strain of ZIKV in a lethal challenge model, with no detectable challenge virus in the female reproductive tract, providing complete protection to the fetuses of these mice. The research group also observed that when administered together but as distinct candidates, the VM-NS1 and VM-E2 vectors induced long-lived plasma cell responses, suggesting that NS1 antibodies may play a potential enhancing role in the protection conferred by ZIKV-E antibodies in the female reproductive system [162].

#### 4.6.2. RhAd52-prM-Env

Three distinct vaccine platforms against ZIKV infection were developed by the Beth Israel Deaconess Medical Center (BIDMC), with vaccine trials conducted in Rhesus monkeys. The vaccine platforms included a purified, inactivated virus vaccine, and two others capable of expressing the envelope and pre-membrane proteins of ZIKV: one in plasmid DNA form and the other as a vectorized Rhesus adenovirus serotype 52. The inactivated virus vaccine provided complete protection to monkeys challenged with ZIKV strains from Puerto Rico and Brazil, also conferring passive protection in adoptive transfer assays in mice. The other two candidates were also capable of completely protecting monkeys challenged with ZIKV, leading to the production of neutralizing antibodies [151].

#### 4.6.3. AdC7-M/E

This vaccine alternative was produced based on a recombinant chimpanzee adenovirus type 7 (AdC7) capable of expressing the M/E glycoproteins of ZIKV. Trials conducted in mice indicated that a single dose was sufficient to generate protective immunity and potent neutralizing antibodies against ZIKV 1 week after vaccination, eliminating the viral load and viremia in all evaluated tissues. The AdC7-M/E candidate also provided protection against testicular damage induced by ZIKV infection. Overall, AdC7-M/E showed promise for future vaccine trials [163].

### 4.7. Zika Virus-like Particles (VLPs)

The strategy developed by TechnoVax involved the production and assembly of virus-like particles (VLPs) co-expressing the non-structural proteins NS2B/NS3 (plasmid ZO3) and structural proteins CprME (plasmid ZO2) of the ZIKV. As a reference for comparison, the researchers used compositions of an inactivated ZIKV (In-ZIKV) for immunogenicity tests conducted in mice. Overall, mice immunized with VLPs showed higher titers of neutralizing antibodies than the formulations of the In-ZIKV vaccine. Additionally, the data demonstrated that higher doses of VLPs combined with adjuvants potentiated the induced serum neutralizing activity. The serum activity generated by the VLP vaccine reached titers higher than those observed in a Brazilian patient infected with the virus. Through comparisons of the neutralization generated by VLPs and In-ZIKV, it was observed that chemical inactivation is deleterious to the neutralization of E protein epitopes, making the VLP vaccine a preferable candidate in subsequent trials [164].

## 5. Challenges and Perspectives

The present review described the state of the art of vaccine development against the most important urban epidemic arboviruses: DENV, CHIKV, and ZIKV (Table 1). Vaccines are the mainstay for the prevention of infectious diseases. It is undeniable that one of the major scientific advancements of the 21st century is the development of vaccines against diseases that cause mortality and morbidity, bringing direct benefits in economic and public health terms. Nevertheless, vaccine development and deployment involve several factors that can directly impact its success [165]. From initial conception to regulatory approval, vaccine candidates undergo multiple rigorous stages, including proof of concept, clinical trials in different phases, and approval by regulatory agencies [166].

The development of vaccines against DENV, CHIKV, and ZIKV is hindered by challenges such as genetic diversity inherent to these RNA viruses, which could create difficulties in creating a vaccine effective against multiple strains circulating in different parts of the world. Another factor relevant to vaccine development is the role of T cells in protection against disease. Future clinical studies are essential to precisely determine the role of T cells in providing protection against these arbovirus infections in humans. Enhancing the understanding and stimulation of strong T-cell immunity may be essential for achieving effective and durable protection against these arboviruses.

In addition to the challenges inherent in vaccine development and evaluation, large-scale production and distribution logistics are critical factors. During the COVID-19 pandemic, for example, distribution to remote regions was hindered by difficulties in maintaining proper storage conditions during transport, leading to dose wastage. Concurrently, the development of these vaccines can also accentuate disparities in access, use, and the likelihood of infection depending on the specific region, indicating that vaccine development alone is just one of several needs in the context of public health.

These arboviruses share vectors with high adaptive capacity in diverse environments. Furthermore, the lack of good sanitation infrastructure provides a potential breeding ground for these mosquitoes [167] (Figure 5).

The majority of inhabited areas on the planet still exhibit deficiencies in sanitation (Figure 6), highlighting how infrastructure issues expose populations in diverse areas of the world to these infections, hindering vector control and, consequently, increasing the transmission of these arboviruses. Vaccination coupled with the availability of clean water and sanitation are public health solutions that yield global-level results [165], demonstrating the direct relationship between these factors. These diseases mainly affect the poor population, which does not represent a profitable market for vaccine companies for final customer selling of the products. Therefore, the participation of governments is imperative to support the development and/or the purchase of these vaccines for distribution to the population at risk.

Despite being the most economical and effective treatment in combating infectious diseases, multi-dose vaccine options result in sub-immunization due to the need for recurring vaccination schedules. Single-dose vaccines based on controlled antigen release systems can address this issue, but their development is inherently challenging [169]. These arboviruses have highly distinctive molecular contexts, requiring their vaccines to encompass different serotypes, lineages, and strains, intensifying the challenge. A multivalent vaccine including the four serotypes of dengue, chikungunya, and Zika is a strategic objective for public health. This combination vaccine will lead to better adherence to immunization schedules and reduce distress to the recipients and vaccination costs.

Moreover, some issues go beyond the boundaries of biological challenges and social neglect. Even after its approval, the acceptance of a vaccine is also influenced by public opinion and politics. The possibility of refusal or delay in accepting safe vaccines has been attributed to the term “vaccine hesitancy”, as was the case with the world’s first dengue vaccine capable of generating immunogenicity against all four serotypes of the disease, Dengvaxia [170]. Furthermore, the episode in question led the country to face vaccination refusals by Filipino parents, even against infections preventable by vaccination, such as measles.

The main need in problematic contexts like this is to separate the narrative regarding other vaccines, focusing the attention and promotion on vaccination programs for available vaccines that have fulfilled their role with the necessary efficacy and safety [170], as in the case of SARS-CoV-2 and various other examples throughout history; the landscape is undeniably challenging, with various factors intensifying the struggle. Successful vaccination depends directly on the mutual contribution between the scientific community and the government, maximizing its impact through improvements in the supply and distribution of vaccines and information.

## Figures and Tables

**Figure 1 viruses-17-00382-f001:**
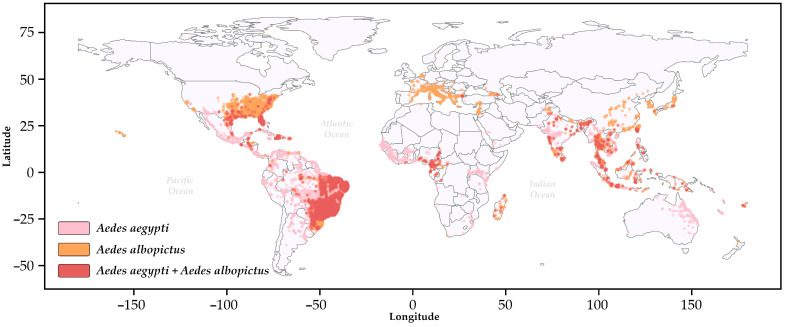
The global distribution of *Aedes aegypti* (pink) and *Aedes albopictus* (sandy brown) based on the VectorMap repository. The red (Indian red) dots indicate overlapping occurrences, with country boundaries shown as solid black lines. The colors used to designate the distribution of the vectors were obtained from the open-source library matplotlib (https://matplotlib.org/, accessed on 19 February 2025) in Python.3.12.2. Additional information is provided in the Appendix A [15,16].

**Figure 2 viruses-17-00382-f002:**
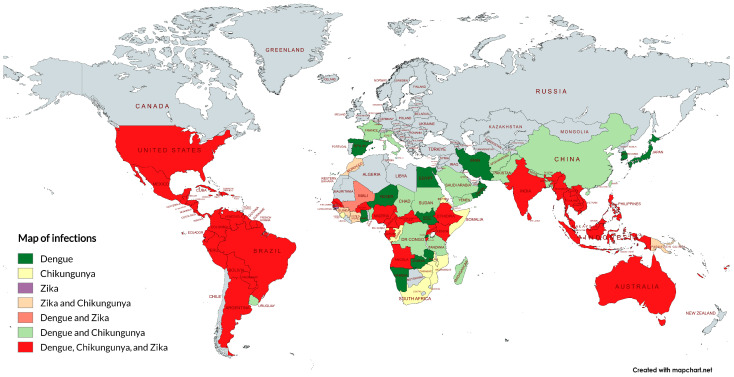
Autochthonous transmission identified for dengue, chikungunya, and Zika viruses and their respective overlap areas. The map was generated using the free tool MapChart 5.11.0 [34] and constructed based on the literature and government reports [35,36,37,38,39,40]. Note that the prevalence is not uniform within each country and may vary according to factors such as climate, geography, and the region’s vector ecology.

**Figure 3 viruses-17-00382-f003:**
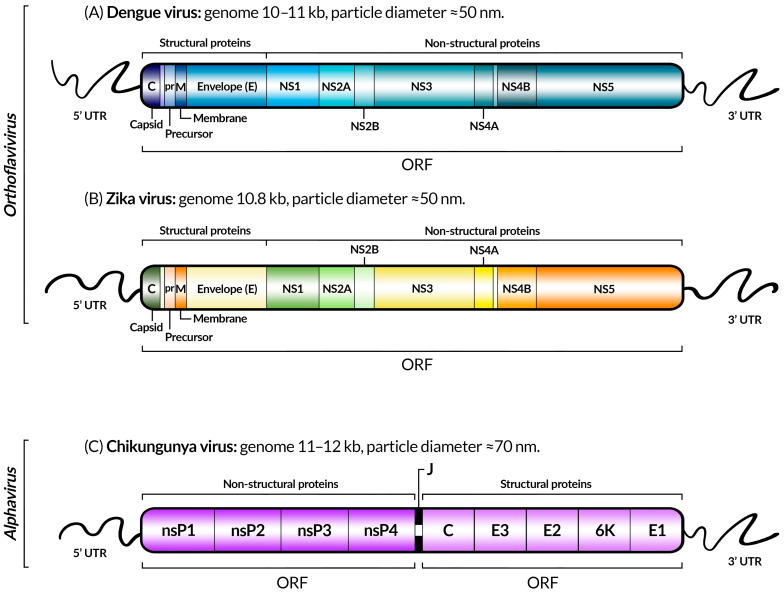
Schematic representation of the genomes of dengue (**A**), Zika (**B**), and chikungunya (**C**) viruses, indicating coding genes, untranslated regions (UTRs), open reading frames (ORFs), genome sizes, the taxonomic genus, and approximate diameter of the viral particles of each virus, as described in [54,55,56] for dengue, Zika, and chikungunya, respectively. The gene colors were chosen for visual purposes, and the difference is not significant. Between the ORFs of CHIKV, there is another UTR sequence: the junction region (J).

**Figure 4 viruses-17-00382-f004:**
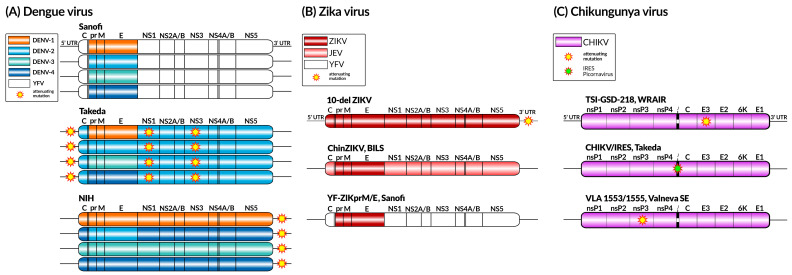
Live attenuated vaccine constructs for dengue (**A**), Zika (**B**), and chikungunya (**C**) viruses. The genomic components of each virus are represented using colors, along with the presence and location of known attenuating mutations. YFV—yellow fever virus; JEV—Japanese encephalitis virus; IRES—Internal Ribosome Entry Site.

**Figure 5 viruses-17-00382-f005:**
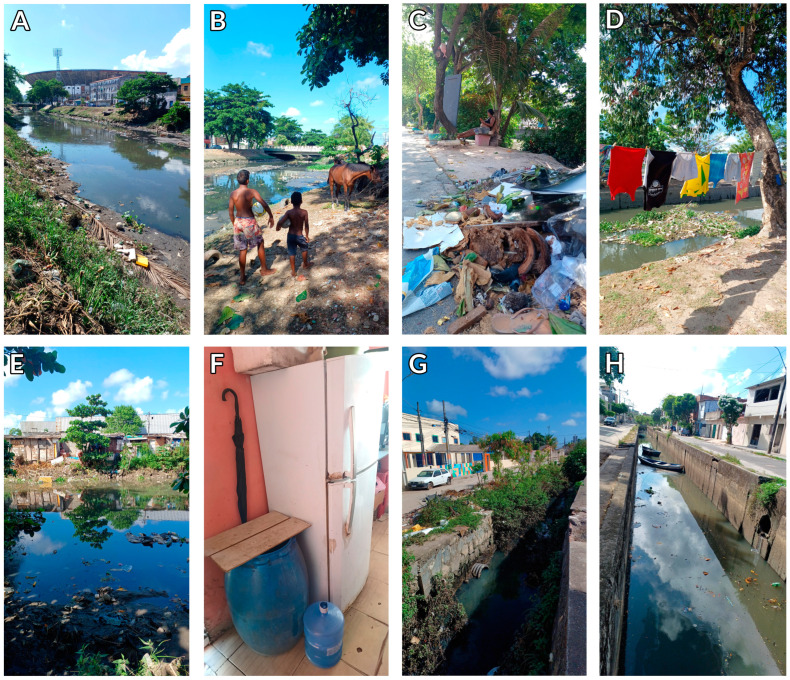
Poor sanitation and housing conditions typically found in the Brazilian big cities’ outskirts. The images are from Recife, a To3.7-million-person metropolitan city located in Northeast Brazil. (**A**) A canal heavily contaminated with sewage and trash near a 60,000-seat soccer stadium; (**B**) young boys and horses that live near the canal; (**C**) a five-month pregnant woman resting on the canal bank just a few meters from a pile of trash; (**D**) clothesline drying on the canal bank; (**E**) housing conditions in a slum (favela in Portuguese) located on the canal bank; (**F**) water containers used by a nearby restaurant to store water; (**G**) open sewage and trash in front of a kindergarten (colored building); (**H**) a canal heavily contaminated with untreated sewage and boats used for navigation by local people.

**Figure 6 viruses-17-00382-f006:**
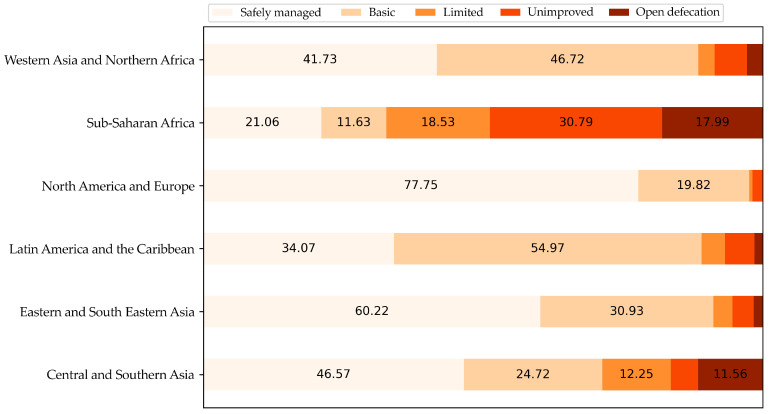
Percentage of people across the world with access to safely managed sanitation (2020). Data source: WHO/UNICEF Joint Monitoring Programme (JMP) for Water Supply and Sanitation (processed by Our World in Data) [168].

**Table 1 viruses-17-00382-t001:** Information on vaccines and vaccine candidates at various stages of development.

Virus	Strategy	Candidate Name	Sponsor	Antigen	Phase Actual	Registers
DENV	Live attenuated virus	CYD-TDV Dengvaxia	Sanofi	Proteins prM and E replaced by the corresponding genes of the four wild serotypes of dengue in the YF17D strain (yellow fever)	Phase 3	NCT01374516 NCT01373281 NCT00842530
		Takeda’s QDENGA	Takeda Pharmaceutical	DENV-2 as the genetic backbone for the four serotypes	Phase 2	NCT02747927 NCT02302066
		LATV TV003 and TV005	NIAID and Butantan Institute	DENV I, III, and IV have the complete genome; DENV II has prM and E, with the rest of the genome from DENV IV	Phase 3	NCT02406729 NCT01696422 NCT02332733 NCT02678455 NCT02879266 NCT02873260 NCT02317900 NCT03416036
		TLAV	NMRC and USAMRMC	Tetravalent live attenuated virus	-	-
	Subunit	V180	Merck Sharp & Dohme LLC	Truncated viral envelope proteins (DEN-80E)	Phase 1	NCT01477580
	Inactivated virus	DPIV	USAMRDC	Tetravalent live attenuated virus	Phase 1	NCT01702857
		TPIV	USAMRDC	Tetravalent live attenuated virus	-	-
	DNA	D1ME100	USAMRDC	prM and E proteins	Phase 1	NCT00290147
		pNS1/E/D2	CNPq, FAPERJ, INCTV, CAPES, and FIOCRUZ	prM and E proteins with adjuvant	-	-
		TVDV	WRAIR	prM and E proteins with adjuvant	-	-
	Live virus	DENV2/4EDll	Baric Lab, De Silva Lab, and NIH	Chimeric virus with the EDIII domain of the E glycoprotein from DENV2 replaced by the EDIII of DENV4	-	-
CHIKV	Live attenuated virus	TSI-GSD-218	USAMRIID/WRAIR	Complete virus with mutations in the E2 glycoprotein	Phase 2	-
		CHIKV/IRES	Takeda Pharmaceutical	Complete virus with insertion of the picornavirus Internal Ribosome Entry Site (IRES) into the CHIKV genome	Phase I	-
		VLA1553	Valneva SE	Complete virus with a mutation in the nsP3 gene	Phase 3	NCT04546724 NCT04838444
		VLA1555	Valneva SE	Complete virus with a mutation in the nsP3 gene	Phase 3	NCT04786444
	Complete inactivated virus	BBV87	Bharat Biotech International Limited		Phase 3	NCT04566484
	Virus-like particles (VLPs)	PXVX0317	Emergent BioSolutions	Structural genes of CHIKV strain 37997	Phase 3	NCT03483961 NCT05072080 NCT05349617
	Viral vector	MV-CHIKV	Themis Biosciences and MSD	Structural proteins of CHIKV vectored from the measles virus	Phase 2	NCT03028441 NCT02861586 NCT03101111
		ChAdOx1-CHIK	University of Oxford	Complete structural polyprotein of CHIKV (capsid, E3, E2, 6K, and E1)	Phase 1	NCT03590392
		VSVΔG-CHIKV	-	Complete envelope polyprotein of CHIKV (E3, E2, 6K, and E1)	-	-
		MVA-CHIK	NIH	E3, E2, and NS1 proteins of CHIKV	-	-
	mRNA encapsulated in lipid nanoparticle	mRNA-1388	Moderna	-	Phase 1	NCT03325075
ZIKV	Live attenuated virus	10-del ZIKV		Modified live virus with deletion in 3′ UTR region		-
		ChinZIKV	BILS	Chimeric virus with Zika prM and E genes in a JEV backbone	-	-
		YF-ZIKprM/E	Sanofi	17D YFV vaccine expressing prM/E from ZIKV	-	-
	Complete inactivated virus	ZPIV	WRAIR/BIDMC	Inactivated virus with adjuvant	Phase 1	NCT02963909 NCT02952833 NCT02937233 NCT03008122
		PIZV/TAV-426	Takeda	Inactivated virus with adjuvant	Phase 1	NCT03343626
		Zika virus vaccine MR766	Bharat Biotech International	Inactivated virus with adjuvant	-	-
	DNA	GLS-5700	GeneOne Life Science/Inovio Pharmaceuticals	-	Phase 1	NCT02809443 NCT02887482
		VR5283	VRC/NIAID	prM-E of Zika virus and Japanese encephalitis virus (JEV) chimera	Phase 1	NCT02840487
		VRC5288	VRC/NIAID	prM-E Polynesian strain inserted into the JEV genome	Phase 2	NCT02996461 NCT03110770
	mRNA	mRNA-1325	Moderna	mRNA encapsulated in nanoparticles expressing prM-E glycoproteins of Zika	Phase 1	NCT03014089
		mRNA–LNP	University of Pennsylvania	mRNA encapsulated in nanoparticles expressing prM-E glycoproteins	Phase 1	NCT02996890
		PzE	Arizona State University	ZIKV E protein expressed in tobacco plants (*Nicotiana benthamiana*)	-	-
		AGS-v	NIH	Synthetic peptides derived from mosquito salivary proteins	Phase 1	NCT03055000
	Protein	E protein from insect S2 cells	University of Hawaii	ZIKV E protein expressed in *Drosophila melanogaster* cells	-	-
	Viral vector	MV-Zika	Themis Bioscience	Measles virus (MV) modified to express the E and NS1 proteins of ZIKV	Phase 1	NCT02996890
		RhAd52-prM-Env	BIDMC	Vector derived from the Rhesus monkey adenovirus (RhAd52) to express the prM and E ZIKV proteins	-	-
		AdC7-M/E	BILS	Recombinant chimpanzee adenovirus for expressing prM-E proteins of ZIKV	-	-
	VLP	Zika virus-like particles	TechnoVax	Virus-like particles co-expressing NS2B/NS3 and CprME proteins	-	-

NIAID—National Institute of Allergy and Infectious Diseases; NMRC—Naval Medical Research Center; USAMRMC—U.S. Army Medical Research and Materiel Command; USAMRDC—U.S. Army Medical Research and Development Command; CNPq—National Council for Scientific and Technological Development; FAPERJ—Carlos Chagas Filho Foundation for Research Support in the State of Rio de Janeiro; INCTV—National Institute of Science and Technology in Vaccines; CAPES—Coordination for the Improvement of Higher Education Personnel; FIOCRUZ—Institute Oswaldo Cruz; WRAIR—Walter Reed Army Institute of Research; NIH—National Institutes of Health; USAMRIID—U.S. Army Medical Research Institute of Infectious Diseases; BIDMC—Beth Israel Deaconess Medical Center; VRC—Vaccine Research Center; NIAID—National Institutes of Allergy and Infectious Diseases; BILS—Beijing Institutes of Life Science, Chinese Academy of Sciences.

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
