# Peer review of "Vaccines Against Urban Epidemic Arboviruses: The State of the Art"

_viruses, 2025, doi:10.3390/v17030382_

Round 1
Reviewer 1 Report
Comments and Suggestions for Authors
I commend the authors for taking on the large task of providing an updated review on the status of the vaccines for these three arboviruses.
Overall the text should be updated to better reflect a central thesis, which seems to be a summary of vaccines against the three viruses. There is an introduction, brief summary of each virus, extensive referencing and survey of vaccine candidates (from mouse studies to vaccines that obtained approval), and then a challenges section detailing poor sanitation conditions. The article seems imbalanced and without a central focus. For example, there is a passage with only a reference from a book chapter detailing a very specific clinical development workflow (Lines 76-85). This paragraph does not fit into the rest of the manuscript because the rest of the paper does not really explain how the arduous clinical development process hindered any of the vaccines described.
Also, the presentation of the data is lacking. The state of Alaska in the United States of America is not at risk for dengue, yet the map has this state colored in as an area of transmission, presumably because it is part of the country, USA.
It is unclear what the basis is for the differentiation of the genome organization of dengue and Zika viruses. For example, both viruses have a “pr” peptide that requires cleavage for complete virion maturation.
The entire manuscript requires extensive editing. For example, the map colors do not match the names of the colors in the text (line 43). Language translation is needed for common usage and short words “e” instead of “and”. Lines 55-56: This statement should be rewritten to more accurately represent that not every tropical country has a triple virus epidemic and that clinical signs cannot be distinguished. Perhaps sometimes they cannot be distinguished or they may be difficult to distinguish. Another example is the strong assertion that coinfections occur from the same mosquito while providing only a single reference of co-infection into mice. Again, this seems like an exaggeration that should be rewritten to more accurately state that experimental data combined with clinical data (need reference of coinfection in humans) indicate one mosquito can transmit two viruses. However, the frequency with which this may occur is poorly understood.
Additional examples of editing needed:
- There are typographical errors throughout. Example) “CYD-TDV” appears as CYT-TDV in some places in the manuscript. In the vaccine summary table, the the description of the antigen in LATV TV003/TV005 is incorrect.
- Line 132-134DENV is an enveloped virus with approximately 50 nm virus with a single strand 132 RNA virus with positive-sense genome; which means it can be directly translated into 133 proteins [34].
- Line 217-218 The results bring an acceptable vaccine safeness to dengue-naive individuals and also 217 dengue previous exposed subjects [69].
The manuscript includes a comprehensive survey of literature and trials, and there is a lot of potential value to produce a great article that truly updates the vaccine efforts to date or focuses on the challenges that prevent the most successful candidates from progressing. Making the decision to focus on one or the other would strengthen the manuscript. Another suggestion is to provide better granularity regarding the current relevance of each vaccine described. For example, Section 3.4.3 describes and example of a vaccine candidate in mice, yet, it is not clear why that study is summarized and not other mouse studies. Additionally, some of the vaccine studies are decades old, and by not addressing this fact, there is a false equivalence of the various preclinical and clinical candidates.
Comments on the Quality of English LanguageSee comments above.
Author Response
Reviewer 1:
I commend the authors for taking on the large task of providing an updated review on the status of the vaccines for these three arboviruses.
- Overall the text should be updated to better reflect a central thesis, which seems to be a summary of vaccines against the three viruses. There is an introduction, brief summary of each virus, extensive referencing and survey of vaccine candidates (from mouse studies to vaccines that obtained approval), and then a challenges section detailing poor sanitation conditions.
- The article seems imbalanced and without a central focus. For example, there is a passage with only a reference from a book chapter detailing a very specific clinical development workflow (Lines 76-85). This paragraph does not fit into the rest of the manuscript because the rest of the paper does not really explain how the arduous clinical development process hindered any of the vaccines described.
Response: Thank you for the comment. The manuscript has been completely revised to better express its main focus: an updated summary of the vaccines developed against dengue, chikungunya, and Zika virus. We fully agree with the reviewer´s observation, and therefore, we have removed the paragraph. The modifications can be found on page 3, lines 98-107.
- Also, the presentation of the data is lacking. The state of Alaska in the United States of America is not at risk for dengue, yet the map has this state colored in as an area of transmission, presumably because it is part of the country, USA.
Response: Thank you for the comment. The image has been updated to correctly reflect that Alaska is not part of the infection area for dengue, Zika, or chikungunya. We have also removed continental Chile, since the arbovirus cases were reported only in Easter Island, but not in the continent. We have also updated the legend of figure 2 to make it clear that the prevalence is not uniform within each country, and may vary according to factors such as climate, geography, and the region’s vector ecology. The modifications are shown on page 3, in the legend of Figure 2, lines 95-97.
- It is unclear what the basis is for the differentiation of the genome organization of dengue and Zika viruses. For example, both viruses have a “pr” peptide that requires cleavage for complete virion maturation.
Response: Thanks for the observation. We fully agree with the reviewer. Figure 3 has been corrected to accurately reflect the genomic organization of the three mentioned arboviruses. Additionally, details that were not previously included have been added, such as the approximate diameter of the viral particle for each virus and the genus to which they belong. The background color of the image was also removed to improve visibility, focusing solely on the genomes and other information present in the figure. The legend has also been updated, indicating the new information present in the image and the references used as the basis for each of the represented genomes. The modifications are shown on page 13, in Figure 3 and its legend, lines 169-172.
- The entire manuscript requires extensive editing. For example, the map colors do not match the names of the colors in the text (line 43).
Response: Thanks for the insight. The manuscript has been thoroughly edited to assure accuracy and improved readability. The description of the colors according to the vector, as well as the overlapping areas, has been corrected. Now, the legend includes the exact name of each color used and the library from which these colors were retrieved. The modifications are shown on page 2, in legend of Figure 1, lines 47-51.
- Language translation is needed for common usage and short words “e” instead of “and”.
Response: Thank you for the comment. The correction has been applied, and the text has been updated in several points, replacing the connective “e” to “and” in the sections that were previously not translated into English. The modifications are shown on page 5-7, Table 1.
- Lines 55-56: This statement should be rewritten to more accurately represent that not every tropical country has a triple virus epidemic and that clinical signs cannot be distinguished. Perhaps sometimes they cannot be distinguished or they may be difficult to distinguish.
Response: Thank you for the timely comment. The paragraph has been rewritten to clearly represent regions with records of dual or triple infections. The difficulty in diagnosing and identifying multiple infections has also been mentioned. The modifications are shown on page 2-3, lines 69-82.
- Another example is the strong assertion that coinfections occur from the same mosquito while providing only a single reference of co-infection into mice. Again, this seems like an exaggeration that should be rewritten to more accurately state that experimental data combined with clinical data (need reference of coinfection in humans) indicate one mosquito can transmit two viruses. However, the frequency with which this may occur is poorly understood.
Response: Thank you once again for the relevant comment. The information has been rewritten and corrected. We have added references describing the coinfections of these arbovirus in humans, including the review by Vogels et al., 2019 (https://doi.org/10.1371/journal.pbio.3000130). The modifications can be observed on page 2, lines 69-82.
- There are typographical errors throughout. Example) “CYD-TDV” appears as CYT-TDV in some places in the manuscript. In the vaccine summary table, the description of the antigen in LATV TV003/TV005 is incorrect.
Response: Thank you for your comment. The manuscript has been thoroughly edited to assure accuracy and improved readability. The modifications can be found on page 5, Table 1; page 14, lines 213 and 221.
- Line 132-134 DENV is an enveloped virus with approximately 50 nm virus with a single strand 132 RNA virus with positive-sense genome; which means it can be directly translated into 133 proteins [34].
Response: Thank you for the comment. This part of the sentence has been corrected and rewritten for better understanding. The modifications can be found on page 12, lines 160-162.
- Line 217-218 The results bring an acceptable vaccine safeness to dengue-naive individuals and also 217 dengue previous exposed subjects [69].
The manuscript includes a comprehensive survey of literature and trials, and there is a lot of potential value to produce a great article that truly updates the vaccine efforts to date or focuses on the challenges that prevent the most successful candidates from progressing. Making the decision to focus on one or the other would strengthen the manuscript. Another suggestion is to provide better granularity regarding the current relevance of each vaccine described. For example, Section 3.4.3 describes and example of a vaccine candidate in mice, yet, it is not clear why that study is summarized and not other mouse studies. Additionally, some of the vaccine studies are decades old, and by not addressing this fact, there is a false equivalence of the various preclinical and clinical candidates.
Response: The reviewer raised an interesting point. We opted to present the different vaccine platforms against each virus and discuss the more advanced studies done in humans. When not available, we included pre-clinical studies in mice and other animal models. The systematical inclusion of pre-clinical studies would make this review too lengthy and cumbersome. However, we have rewritten the Perspective section to address some of the challenges that prevent the most successful candidates from progressing. The modifications can be found on page 31-34.
Comments on the Quality of English Language
See comments above.
Response: Thanks for the insight. The manuscript has been thoroughly edited to assure accuracy and improved readability.
Reviewer 2 Report
Comments and Suggestions for Authors
In this current manuscript, the authors provide a thorough and timely review of vaccine candidates targeting dengue, chikungunya, and Zika viruses. I commend the authors for the breadth and depth of their coverage of vaccine platforms being tested for each virus. I also commend the authors for their focus on the issues and challenges regarding the production and deployment of these vaccines to populations at risk. My comments are minor:
line 94: I recommend use of "preexistent technology" rather than "used technology" as the former is clearer, more understandable.
line 398-408: the text should not be italicized
line 673: I recommend adding mention of two recent studies which demonstrate the safety and efficacy of ZPIV during pregnancy using a marmoset infection model (PMID 38360793, PMID 38368443). Importantly, data show that vertical transmission of ZIKV could be prevented by pre-pregnancy vaccination.
Author Response
Reviewer #2:
In this current manuscript, the authors provide a thorough and timely review of vaccine candidates targeting dengue, chikungunya, and Zika viruses. I commend the authors for the breadth and depth of their coverage of vaccine platforms being tested for each virus. I also commend the authors for their focus on the issues and challenges regarding the production and deployment of these vaccines to populations at risk. My comments are minor:
Response: We appreciate the positive feedback from the reviewer.
line 94: I recommend use of "preexistent technology" rather than "used technology" as the former is clearer, more understandable
Response: Thank you for the comment. We agree with the suggestion. The modification has been made and can be found on page 4, line 117.
line 398-408: the text should not be italicized
Response: Thanks for the comment. The modification has been made and can be found on page 21, lines 445-462.
line 673: I recommend adding mention of two recent studies which demonstrate the safety and efficacy of ZPIV during pregnancy using a marmoset infection model (PMID 38360793, PMID 38368443). Importantly, data show that vertical transmission of ZIKV could be prevented by pre-pregnancy vaccination.
Response: Once again, thank you for your comment. We agree with the relevance of the cited articles, and both have been added in the ZPIV vaccine section. The modifications can be found on page 27-28, lines 778-787.
Reviewer 3 Report
Comments and Suggestions for Authors
This paper by de Moura Pereira and colleagues presents an updated review on the current state-of-the-art on vaccine candidates against the urban epidemic arboviruses dengue (DENV), chikungunya (CHIKV), and Zika (ZIKV) viruses, including discussions about the challenges related to their production and deployment within relevant geographic regions. The topic of the review is important and timely.
For each of the arboviruses (DENV, CHIKV, and ZIKV), the authors have provided a concise but comprehensive and clear summary about their epidemiology, disease manifestations and their impact in public health, followed by a summary of the existing candidate vaccine platforms and their status regarding their performances in preclinical studies and clinical trials.
The review is for the most clearly written and well organized and provides interested non-expert investigators with a good introduction to this area of research.
Table 1 and figures 1-3, and 5 are informative and complement well the text, whereas figure 4 does not seem to provide any useful information.
There are several issues that need additional elaboration or clarification:
1) The presentation of the different vaccine platforms has included a rather uneven discussion of the contribution of T cell responses to protection, specifically their roles in protection from severe disease, and whether additional antigens should be considered for incorporation into vaccine candidates.
2) For the live-attenuated vaccine platforms, it would be helpful to incorporate some comments about what is known about the mechanisms of attenuation and their genetic and phenotypic stability, as well as whether there are any concerns of reversion to a virulent form of the corresponding virus.
3) It would be helpful for CHIKV and ZIKV to incorporate a brief discussion about whether viral genetic diversity, and the virus potential for rapid evolution, can pose problems for an effective durable vaccine.
4) The description of the VSV∆G-CHIKV platform is rather unclear, and it is recommended to rewrite this section to improve its clarity.
5) The consideration of mRNA-1944 as a CHIKV vaccine is questionable, as this reagent fits better the definition of an immune therapeutic molecule.
6) Section 4.1.3 describing the YVF-17D based ZIKV live-attenuated vaccine candidate should specify whether the same degree of protection was observed in the different immunocompromised mouse strains.
7) The authors should provide some basic information about the backbone of vectors VRC5283 and VRC5288.
8) Discussing costs and logistics, including preservation of biological activity and delivery, associated with the different vaccine candidates.
9) The review would benefit from the incorporation of table including basic information comparing the different platforms for each of the arboviruses.
Comments on the Quality of English LanguageN/A
Author Response
Reviewer #3:
This paper by de Moura Pereira and colleagues presents an updated review on the current state-of-the-art on vaccine candidates against the urban epidemic arboviruses dengue (DENV), chikungunya (CHIKV), and Zika (ZIKV) viruses, including discussions about the challenges related to their production and deployment within relevant geographic regions. The topic of the review is important and timely.
For each of the arboviruses (DENV, CHIKV, and ZIKV), the authors have provided a concise but comprehensive and clear summary about their epidemiology, disease manifestations and their impact in public health, followed by a summary of the existing candidate vaccine platforms and their status regarding their performances in preclinical studies and clinical trials.
The review is for the most clearly written and well organized and provides interested non-expert investigators with a good introduction to this area of research.
Response: We appreciate the positive feedback from the reviewer. Indeed, it was a herculean effort to put together the most relevant literature about the vaccines against these three important arboviruses in a concise way.
Table 1 and figures 1-3, and 5 are informative and complement well the text, whereas figure 4 does not seem to provide any useful information.
Response: We understand the point raised by the reviewer. Nevertheless, we strongly believe that figure 5 (former figure 4) strikingly represents the ecoepidemiological settings where these arboviruses circulate and cause disease. The pictures were taken in a large metropolitan city in Brazil, which lacks proper sanitation and represents well the scenario seen in most of tropical countries.
There are several issues that need additional elaboration or clarification:
1) The presentation of the different vaccine platforms has included a rather uneven discussion of the contribution of T cell responses to protection, specifically their roles in protection from severe disease, and whether additional antigens should be considered for incorporation into vaccine candidates.
Response: The reviewer raised an interesting point. Indeed, enhancing the understanding and stimulation of strong T-cell immunity may be essential for achieving effective and durable protection against these arboviruses. We opted not to discuss this matter deeply in the review as this would make it too lengthy and cumbersome. However, we have added a paragraph in the perspective to address this question. The modifications can be found on page 31, lines 958-962.
2) For the live-attenuated vaccine platforms, it would be helpful to incorporate some comments about what is known about the mechanisms of attenuation and their genetic and phenotypic stability, as well as whether there are any concerns of reversion to a virulent form of the corresponding virus.
Response: Thank you very much for the comment. We have updated Table 1, including additional information about the vaccines. We have also made a new figure (figure 4) illustrating the main live-attenuated vaccine platforms developed so far and their attenuation mechanism. The modifications can be found on page 32.
3) It would be helpful for CHIKV and ZIKV to incorporate a brief discussion about whether viral genetic diversity, and the virus potential for rapid evolution, can pose problems for an effective durable vaccine.
Response: Thank you very much for the comment. We have added a brief discussion about the implications of the arbovirus genetic diversity for vaccine development. The modifications can be found on page 31, lines 955-958.
4) The description of the VSV∆G-CHIKV platform is rather unclear, and it is recommended to rewrite this section to improve its clarity.
Response: Thank you very much for the comment. This section has been completely rewritten to clarify the information related to the VSVΔG-CHIKV vaccine. The modifications can be found on page 24, lines 604-621.
5) The consideration of mRNA-1944 as a CHIKV vaccine is questionable, as this reagent fits better the definition of an immune therapeutic molecule.
Response: Thank you once again for the pertinent comment. We agree with the observation and have removed the section from the text. It is correct to categorize mRNA-1944 as a therapeutic molecule rather than a vaccine, as its direct function is to modulate the immune system and stimulate an immune response against CHIKV, rather than inducing, for example, long-term immunity through the generation of memory cells, a key feature of a typical vaccine. The modifications can be found on page 25, lines 654-669.
6) Section 4.1.3 describing the YVF-17D based ZIKV live-attenuated vaccine candidate should specify whether the same degree of protection was observed in the different immunocompromised mouse strains.
Response: Thank you for the comment. Part of the section has been rewritten to clarify that, regardless of the mouse strain used, the vaccine was effective, providing dual protection against both ZIKV and YFV. The modifications can be found on page 27, lines 751-757.
7) The authors should provide some basic information about the backbone of vectors VRC5283 and VRC5288.
Response: Thanks for the comment. Additional information has been added to the paragraph. The modifications can be seen on the page 28, lines 827-828.
8) Discussing costs and logistics, including preservation of biological activity and delivery, associated with the different vaccine candidates.
Response: Thank you very much for the comment. We have added a brief paragraph about these factors. The modifications can be found on page 31, lines 964-967.
9) The review would benefit from the incorporation of table including basic information comparing the different platforms for each of the arboviruses.
Response: Thank you very much for the comment. We have updated Table 1, including additional information about the vaccines. We have also made a new figure (figure 4) illustrating the main live-attenuated vaccine platforms developed so far and their attenuation mechanism. The modifications can be found on page 32.